# Preventing postoperative pulmonary complications by establishing a machine-learning assisted approach (PEPPERMINT): Study protocol for the creation of a risk prediction model

Britta Trautwein[1]*, Meinrad Beer[1], Manfred Blobner[1,2], Bettina Jungwirth[1], Simone Maria Kagerbauer[1,2], Michael Götz[1]

**1** University Hospital Ulm, Ulm, Germany, **2** Technical University of Munich, Munich, Germany

☯ These authors contributed equally to this work.
* britta.trautwein@uniklinik-ulm.de

## Abstract

### Background

Postoperative pulmonary complications (POPC) are common after general anaesthesia and are a major cause of increased morbidity and mortality in surgical patients. However, prevention and treatment methods for POPC that are considered effective tie up human and technical resources. Therefore, the planned research project aims to create a prediction model that enables the reliable identification of high-risk patients immediately after surgery based on a tailored machine learning algorithm.

### Methods

This clinical cohort study will follow the TRIPOD statement for multivariable prediction model development. Development of the prognostic model will require 512 patients undergoing elective surgery under general anaesthesia. Besides the collection of perioperative routine data, standardised lung sonography will be performed postoperatively in the recovery room on each patient. During the postoperative course, patients will be examined in a structured manner on postoperative days 1,3 and 7 to detect POPC. The endpoints determined in this way, together with the clinical and imaging data collected, are then used to train a machine learning model based on neural networks and ensemble methods to predict POPC in the early postoperative phase.

### Discussion

In the perioperative setting, detecting POPC before they become clinically manifest is desirable. This would ensure optimal patient care and resource allocation and help

**Data availability statement:** Data cannot be shared publicly because of Article 9 of the EU General Data Protecion Regulation (GDPR). According to Article 9 of the EU GDPR, health-related data are classified as sensitive personal data, and their processing is generally prohibited unless specific exceptions apply. Additionally, under the German Federal Data Protection Act (Bundesdatenschutzgesetz, BDSG), the processing of health data is subject to strict requirements, particularly regarding pseudonymization and the risk of re-identification. Even with pseudonymization, there remains a risk of re-identification, especially in a dataset obtained from a special patient population in a defined timeframe at a single hospital. Therefore, publishing these data in an open repository would not comply with data protection regulations and could compromise patient confidentiality. For this reason, we are not allowed to publish the raw data publicly. However, we fully agree that aggregated or properly anonymized data, which are no longer considered personal data under the GDPR (Recital 26), can be shared without restrictions, and we will make such data available in a public repository. To promote transparency and reproducibility, we will make the following resources available via a public repository (e.g., GitHub) or as supplemental material of the final publication: • The full source code and model architecture, • A trained version of the predictive model, • A fully anonymized test dataset for evaluation purposes, • Detailed aggregated statistics describing the dataset. Furthermore, for researchers who meet the criteria for access to condifential data, controlled access to the raw data upon reasonable request and in compliance with Article 9 of the GDPR, the BDSG, and local ethical approvals (Ethics committee University of Ulm, mail: ethik-kommission@uni-ulm.de) data will be made available.

**Funding:** The study is funded by the Department of Anaesthesiology and Intensive Care Medicine of the University Hospital Ulm and the associated study centre. The necessary equipment, including ultrasound devices and tablets as well as staffing, is available on site. An intramural funding is used for the resources necessary for data evaluation and model development from the

initiate adequate patient treatment after being transferred from the recovery room to the ward. A reliable prediction algorithm based on machine learning holds great potential to improve postoperative outcomes.

## Trial registration

ClinicalTrials.gov ID: NCT05789953 (29th of March 2023)

## Introduction

The incidence of postoperative pulmonary complications (POPC) varies between 9–40%, depending on the surgical procedure and the definition used [1]. Therefore, the introduction of a standardised definition of the outcome "postoperative pulmonary complications", developed by the Standardised Endpoints for Perioperative Medicine (StEP) collaboration in 2018, represents a prerequisite for future studies in this field [1]. Even supposedly minor complications have the potential to significantly increase the length of hospital stay [2]. Various preoperative risk factors are known but usually cannot be modified. As POPC are the main cause of postoperative morbidity and mortality and the reduction of perioperative mortality is dependent on early recognition and treatment [3], accurate prediction is of paramount interest. There are only a few clinical scoring systems [4]; the currently best-evaluated preoperative score for predicting postoperative pulmonary complications (ARISCAT: Assess Respiratory Risk in Surgical Patients in Catalonia, Table 1) has sufficient sensitivity but lacks specificity [5]. A first retrospective study using machine-learning (ML) methods for determining risk for pneumonia and pulmonary embolism using pre- and intraoperative routine data has shown good accuracy [6]. However, the study shows high specificity but low sensitivity, which could result in overtreatment in clinical practice. In our study, we aim to create a high precision decision-supporting tool for perioperative physicians to identify high-risk patients at an early stage.

However, ML algorithms based only on routine clinical data depend highly on data quality and comprehensiveness. Algorithms based on standardised imaging data are easier to transfer to other facilities and implement in routine clinical practice. Therefore, combining image analysis might be beneficial, especially as sonography is becoming increasingly important as a non-invasive examination method that can be performed at the bedside. Various sonographic scores and models have been developed to predict pulmonary complications [7]. Image processing methods and machine learning, particularly deep learning, are also increasingly used in ultrasound diagnostics [8,9]. Augmented algorithms using pre- and intraoperative clinical information in addition to ultrasound imaging data may provide better predictive accuracy than the respective individual methods. However, to our knowledge, combining routine clinical data and ultrasound imaging data to develop a predictive machine-learning algorithm has not yet been tested. In addition, prospective clinical evaluation of machine-learning algorithm-based prediction models, which is planned herein, lacks to date.

department of radiology, including powerful computers and scientific staff. Furthermore, the study team of the PEPPERMINT study has submitted a grant application to Deutsche Forschungsgemeinschaft (DFG), which is currently being evaluated. The funding will provide financial support only, and has no role in the design, management, analysis, interpretation of the data and reporting of this study. No other institution or industrial company were or will be involved in financing, planning or conducting the study.

**Competing interests:** I have read the journal's policy and the authors of this manuscript have the following competing interests: B. Jungwirth and S. Kagerbauer received grants from Löwenstein Medical Innovation (Berlin, Germany). M. Blobner received research support from MSD (Haar, Germany), fees for consultancy or lectures from GE Healthcare (Helsinki, Finland), Grünenthal (Aachen, Germany), and Senzime (Landshut, Germany). This does not alter our adherence to PLOS ONE policies on sharing data and materials.

**Abbreviations:** ARDS, Acute respiratory distress syndrome; ARISCAT, assess respiratory risk in surgical patients in Catalonia; AI, Artificial intelligence; AUPRC, area under the precision recall curve; AUROC, Area under the receiver operating characteristic; CNN, Convolutional Neural Network; CPAP, Continuous positive airway pressure; CRP, C-reactive protein; DCA, Decision curve analysis; Diast., Diastolic; DICOM, Digital imaging and communications in medicine; DFG, Deutsche Forschungsgemeinschaft; EPCO, European perioperative clinical outcome; FiO2, Fraction of inspired oxygen; ICU, Intensive care unit; ML, Machine learning; NIV, Non-invasive ventilation; PACS, Picture archiving and communication system; PACU, Post-anesthesia care unit; paCO2, Partial pressure of arterial carbon dioxide; paO2, Partial pressure of arterial oxygen; PCT, Procalcitonin; PDMS, Patient data management system; POPC, Postoperative pulmonary complications; SQL, Structured query language; StEP, Standardised endpoints for perioperative medicine; Syst, Systolic;TRIPOD, Transparent reporting of a multivariable prediction model for individual prognosis and diagnosis; QoR-9, Quality of recovery-9

**Table 1. ARISCAT score.**

| Score parameters | components | Risk score |
|---|---|---|
| Age | ≤ 50 years | 0 |
| | 51-80 years | 3 |
| | > 80 years | 16 |
| Preoperative oxygen saturation | ≥ 96% | 0 |
| | 91-95% | 8 |
| | ≤ 90% | 24 |
| Respiratory infection in past 1 month | No | 0 |
| | Yes | 17 |
| Preoperative hemoglobin ≤ 10 mg/dl | No | 0 |
| | Yes | 11 |
| Incision | Peripheral incision | 0 |
| | Upper abdominal incision | 15 |
| | Intrathoracic incision | 24 |
| Surgery duration | <2h | 0 |
| | 2-3h | 16 |
| | > 3h | 23 |
| Emergency procedure | No | 0 |
| | Yes | 8 |
| **ARISCAT Score** | **Risk** | **Total** |
| | Low Risk | < 26 |
| | Intermediate Risk | 26-44 |
| | High Risk | ≥ 45 |

The table was adapted from [5]; The article was published under the Creative Commons Attribution (CC BY 4.0) license (https://creativecommons.org/licenses/by/4.0/). ARISCAT = Assess Respiratory Risk in Surgical Patients in Catalonia.

Measures for preventing POPC, such as postoperative non-invasive ventilation and physiotherapy, are known and considered effective [10,11] but are probably not consistently applied in clinical routine due to the increased demand, especially for human resources.

This study aims to combine pre- and intraoperative data with lung ultrasound imaging in the recovery room to develop an ML-based risk score for POPC. A precise score that reliably identifies patients at risk in the early postoperative phase and simultaneously avoids overtreatment can ensure adequate personalised treatment of postoperative patients.

## Materials and Methods

### Trial registration

Name of the registry: ClinicalTrials.gov
Registration ID: NCT05789953
Approval date: 29/03/2023

## Ethics approval

Name: Ethics committee University of Ulm
Approval Number: 369/22
Approval date: 22/12/2022
Head of committee: Prof. Dr. Florian Steger, Oberberghof 7, 89081 Ulm, Germany
Mail: ethik-kommission@uni-ulm.de
Homepage: https://www.uni-ulm.de/einrichtungen/ethikkommission-der-universitaet-ulm/
Written informed consent to participate will be obtained from all participants.

## Objectives

The main hypothesis of the PEPPERMINT study is that a patient's risk of POPC can be reliably predicted using a machine-learning algorithm and that the predictive accuracy of the algorithm outperforms common clinical scoring systems.

The primary objective is to develop a machine-learning algorithm based on immediately postoperatively obtained lung ultrasound imaging data of adult patients undergoing surgery in general anaesthesia to predict the risk of postoperative pulmonary complications. This model is intended to provide better predictive ability than the currently best-established clinical score, the ARISCAT [5] or a machine learning model solely based on clinical routine data.

The secondary objective is to investigate whether improving model performance by adding clinical routine parameters to the imaging data is possible.

Furthermore, the optimal risk threshold for an intervention will be determined in case of a clinical application of the model.

Further objectives include identifying patient-specific risk factors for POPC through analysis of the collected routine clinical data and modification of the models created to predict the secondary endpoints of hospital length of stay, in-hospital mortality and postoperative quality of recovery.

## Trial design

The PEPPERMINT study is a prospective, single-center clinical cohort study designed to develop and evaluate a risk prediction model for POPC. The study follows the TRIPOD (Transparent Reporting of a Multivariable Prediction Model for Individual Prognosis or Diagnosis) guidelines for multivariable model development and validation [12].

A total of 512 patients will be enrolled based on sample size calculation. For each patient, both clinical data and lung ultrasound images will be prospectively collected. The primary endpoint is the occurrence of any postoperative pulmonary complication, as defined by standardised criteria.

## Three predictive models will be developed

First, a model based on deep learning using lung ultrasound images, second, a model based solely on clinical variables using common frameworks like automated machine learning (AutoML), and finally, a combined model integrating both clinical and imaging features.

The dataset will be split using stratified, patient-wise sampling into non-overlapping training and hold-out test sets. Internal model performance will be assessed using k-fold cross-validation within the training set, and final evaluation will be conducted on the independent test set. Full details of the modelling and validation procedures are provided in the statistical methods.

## Study setting

The PEPPERMINT study will be conducted as a single-centre study at the University Hospital Ulm in Germany. The hospital is a tertiary care and academic hospital where about 30,000 anaesthesia procedures are performed annually, including a broad spectrum of surgical disciplines and interventions. To establish a data processing pipeline for imaging data and to integrate artificial intelligence (AI) algorithms into the study setting, an interface between the ultrasound devices and the hospital´s internal Picture Archiving and Communication System (PACS) has already been established. To bundle the expertise in image processing, collection, and processing of big data, the departments of anaesthesiology and radiology cooperate for this study.

The members of the study group are predominantly physicians and specialists in the fields of anesthesiology and radiology. The leader of the research group "Experimental Radiology", who holds a formal background in engineering, is part of the study team. His expertise lies in machine learning, deep learning, and computer vision, with a focus on medical imaging applications. Another group member holds a master's degree in medical informatics and has substantial experience in machine learning for perioperative risk prediction. The study team is completed by Master's and PhD students from the Department of Radiology with a background in computer science and engineering

In addition to specialised personnel, high computing capacity for fast processing and reliable storage is necessary to develop the model and will be provided by the involved departments.

As the study is designed to build decision-support systems, on-site evaluation is planned as a subsequent step, which will be covered by a separate study. The data is evaluated offline without direct interaction between medical staff and the AI algorithm. Consequently, no feedback loop is planned in this phase.

## Eligibility criteria

Adult patients (≥18 years) of all sexes are eligible for the study.

Patients must meet the following **inclusion criteria**: Scheduling for elective surgical procedures under general anaesthesia with a planned overnight hospital stay, and written informed consent by patients.

If they meet any of the following **exclusion criteria**, they will not be included in the study: Younger than 18 years of age, outpatient surgery, planned postoperative admission to intensive care unit (ICU), need for intensive care treatment before surgery and emergency surgery.

**Secondary exclusion criteria** are: Unplanned hospital discharge/transfer on the day of surgery, which does not allow examination of the primary outcome; cancellation/postponement of index surgery; and immediate unplanned postoperative admission to the ICU due to an intraoperative complication.

Furthermore, the **Inclusion criteria for input data** are: At least two adjacent ribs and the pleura or corresponding pathologies (e.g., pneumothorax, pleural effusion) must be visible on the ultrasound image. The ultrasound examination must cover all 12 defined areas (details in chapter "Study measures").

The **exclusion criteria for input data** are: Ultrasound images on which the leading structures ribs and pleura or alternative pathologies cannot be depicted, or incomplete visualisation of the 12 previously defined examination areas in the thoracic region, e.g., in the case of immobile patients or inaccessibility due to a bandage or drains. Importantly, decisions regarding potential exclusion due to poor image quality will not be made by the examining physician, but rather by an independent, blinded radiologist, thereby minimising the risk of subjective selection bias.

Clinical data is collected from patients throughout their hospitalisation. An employee who is not involved in data collection will carry out a plausibility check and random comparison with the medical records at regular intervals. Implausible data is removed from the data record.

Patients who cannot be visited postoperatively and for whom no endpoint could be defined are excluded from the study; however, their data may be used for secondary analyses. Incomplete pre-operative routine data in the electronic patient record are not a reason for exclusion if the patient can be visited postoperatively

## Recruitement and informed consent

The University Hospital Ulm is a tertiary care hospital where about 30,000 anaesthesia procedures are performed annually. The PEPPERMINT study evaluates a general surgical population; therefore, eligibility criteria were set as low as possible. Approximately 50 patients are seen in the pre-anaesthesia outpatient clinic for preoperative evaluation each working day, of which about 80% will fulfil the inclusion criteria. The abovementioned 3 risk groups, according to the ARISCAT score, will be recruited equally, which means that after about 60 patients in one risk group, it will be paused, and the other risk groups will be prioritized until the same number is present in all risk groups. The equipment and staffing of the recovery rooms and the existing expertise in lung sonography allow the inclusion of 3–5 patients daily so that patient recruitment should be possible without any problems within one year.

Informed consent from trial participants or authorised surrogates will be obtained by a physician from the Department of Anaesthesiology and Intensive Care Medicine of the University Hospital of Ulm. The informed consent discussion will be part of the scheduled informed consent discussion for general anaesthesia. According to the usual routine preoperative procedure, the optimal anaesthesia for the patient is planned based on previous diseases, previous anaesthetic and surgical procedures, and personal preference. In case of doubt, the senior physician will be consulted. If the inclusion criteria are met and the patient gives consent, an informed consent discussion about the PEPPERMINT study will take place. Written informed consent will be obtained from the physicians, who will explain the hospital's policy on data collection and storage, the general process, and the goals of the study. Additional consent for the collection of participant data, which is routinely obtained during standard pre-, intra- and postoperative anesthesiologic care, will be obtained as well.

A participants schedule is shown in Fig 1.

## Study measures

Study-related measures to acquire the imaging data include performing a standardised lung ultrasound on each included patient immediately postoperatively in the recovery room or post-anaesthesia care unit (PACU). Sonographic examination of the lungs is a common, noninvasive bedside procedure. It can be performed without additional positioning in supine position and takes approximately 5 minutes. The examination is performed using a standardised, previously published method [13]. Thereby, a convex probe (5 MHz) and a predefined preset of the ultrasound device (the default "lung" preset of the device) are going to be used. Each hemithorax will be divided into 6 areas, separated by the anterior and posterior axillary lines (anterior, lateral, and posterior) and a superior and inferior area. For each patient, 12 pictures and 12 videos, one in each area, will be captured. This 12-zone protocol showed the highest intra-class correlation coefficient compared to other protocols [14]. A graphical representation of the areas can be found in Fig 2.

The criteria for including imaging data are described above ("Eligibility Criteria"). If these conditions are not met, the interpretability of the ultrasound is considered insufficient.

To maximize inclusiveness and minimize selection bias, operators are encouraged to adjust technical parameters (e.g., depth and gain) during image acquisition in order to meet these quality criteria. A convex transducer was selected to allow visualization of deeper structures across a wide range of patient anatomies.

The performing physicians are experienced in perioperative medicine and critical care and will be trained in ultrasound methodology before the start of the study.

After transmission and storage in the PACS of the Department of Radiology, image pre-processing and artifact correction are performed before the data serves as input for the machine learning model. Images are not labeled by human experts; only the pre-defined endpoint POPC serves as labels for the imaging data.

The clinical data obtained for the study correspond to parameters routinely collected during anesthesiologic preoperative evaluation, the course of anaesthesia during surgery, and postoperatively in the recovery room. Plausibility checks are carried out on the numerical data; if these deviate from the valid value range, they are removed. Missing data may be imputed by different methods as described below.

| | STUDY PERIOD | | | | |
|---|---|---|---|---|---|
| | Enrolment | Elective surgery | Inpatient hospitalization | | |
| **TIMEPOINT** | | Day 1 | *Day 2* | *Day 4* | *Day 8* |
| **ENROLMENT:** | | | | | |
| **Eligibility screen** | X | | | | |
| **Informed consent** | X | | | | |
| **INTERVENTIONS:** | | | | | |
| *LUS* | | X | | | |
| *Postoperative visit* | | | X | X | X |
| **ASSESSMENTS:** | | | | | |
| *Pre-anaesthesia visit for elective general anaesthesia* | X | | | | |
| *Postoperative, standardised LUS in the recovery room* | | X | | | |
| *Clinical examination and Interview to detect POPC on ward* | | | X | X | X |

**Fig 1. SPIRIT schedule PEPPERMINT trial.** Schematic diagram presenting the schedule for participants, based on SPIRIT schedule; LUS = Lung ultrasound; POPC = postoperative pulmonary complications.

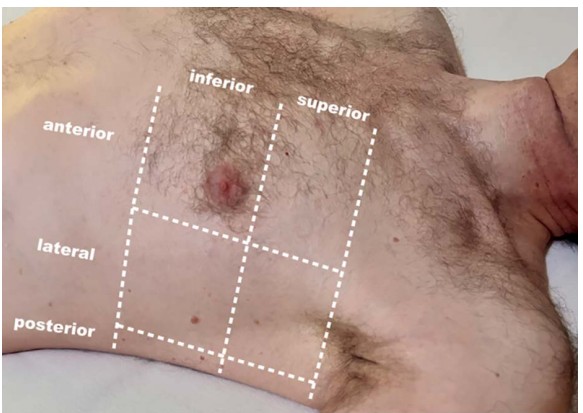

**Fig 2. Graphic representation of the lung ultrasound areas.** Lung ultrasound will be performed in 12 areas, 6 in each hemithorax. Areas are separated by the anterior and posterior axillary line into an anterior, lateral and posterior zone and a superior and inferior area.

Based on the collected data and the outcome parameters (description in chapter "Outcome"), machine learning algorithms will be trained to predict POPC by outputting an individual patient's percentage risk of suffering a postoperative pulmonary complication.

### Modifications, adherence and concomitant care

Regarding the study protocol, the following scenarios allow a change in the study protocol: (1) In case of patient withdrawal of consent, the study protocol will be stopped immediately, and the patient will be excluded from the study. (2) If it is not possible to perform an ultrasound examination that meets the abovementioned quality requirements, the patient will be excluded from the study. (3) Postoperative lung ultrasound will take place in the recovery room; dates might be rescheduled depending on the postponement of the surgery. (4) Postoperative visits on the normal ward will be on postoperative days 1,3 and 7. If the patient is discharged from the hospital or transferred to another hospital within the first 7 days, the study ends on the day of discharge.

To improve workflow, the recruiting anaesthetist in the pre-anaesthesia outpatient clinic receives a simple checklist with eligibility criteria and the ARISCAT score (S1 File), and study information will be handed to patients in the waiting area. To improve adherence to the lung ultrasound protocol in the recovery room, a group of selected anaesthetists will undergo personal training in ultrasound methodology and receive detailed instructions in written form, which are also attached to the ultrasound device. Additionally, pocket cards with brief instructions will be distributed among the responsible physicians.

Postoperative ward visits will be performed by qualified study nurses and trained medical students with a tablet to simplify the visits and enhance data management.

During the trial, the patient will undergo routine perioperative care as per standard. Therefore, the patient receives the concomitant or intervention as per his physicians' decision, and no concomitants or interventions are prohibited during the trial. Relevant information regarding POPC that results from postoperative diagnostics or interventions will be recorded. Supposed complications that have not yet been treated are noticed during the postoperative visit. In that case, the study staff will inform the responsible ward physician to initiate any necessary therapy.

### Sample size

Predicting POPC for the clinician is difficult on a case-by-case basis. Therefore, several scoring systems have been developed in the past. The most common of these, the ARISCAT score, has an AUROC of 0.83 [15]. It should be noted that the current literature does not provide any further metrics such as area under the precision-recall curve (AUPRC) or F1 score on the ARISCAT [15,16]. We therefore used the AUROC as the reference metric. The model we aim to create should be significantly better than the ARISCAT score and thus have at least an AUROC of 0.93. Achieving this seems realistic since, in the preliminary work of our research group, prediction models for various postoperative complications have already been created, whose predictive accuracy is in this range [17,18]. With a significance level of 0.05 and a power of 80%, 512 patients would be needed to create the database based on the method described by Hanley and McNeil for comparing ROC curves [19].

Patients are stratified according to the risk criteria determined by ARISCAT so that approximately equal numbers of patients are included in each of the low-risk (ARISCAT < 26 points), intermediate-risk (ARISCAT 26–44 points), and high-risk (ARISCAT ≥ 45 points) groups.

### Outcomes

The primary outcome of the PEPPERMINT study to be predicted by the machine learning model is the risk of developing postoperative pulmonary complications after surgery in general anaesthesia between postoperative days 1 and 7. POPC will be defined and graded by severity according to the standardised criteria of the StEP collaboration [1]. Complications

not further described by the StEP collaboration will be defined as from the EPCO (European Perioperative Clinical Outcome) task force [20] as listed in Table 2. POPC, as a composite outcome, summarises atelectasis, pneumonia, acute respiratory distress syndrome (ARDS), pulmonary aspiration, pulmonary embolism, pleural effusion, pneumothorax, and bronchospasm [1]. POPC is assumed as soon as at least one of the listed events occurs and will be detected during the postoperative visit or chart review after the patient is discharged.

Outcome assessment will be performed by qualified study staff consisting of three study nurses and two doctoral students on postoperative visits on days 1,3 and 7. To detect complications, visits will include a questionnaire (pulmonary symptoms and mental state), a clinical examination (pulmonary auscultation), the collection of vital parameters (Heart rate, blood pressure, oxygen saturation, breathing rate, temperature) as well as a chart review (oxygen supply, medication, signs of aspiration, admission to ICU). If available during the postoperative course, the following measures will be included: laboratory (c-reactive protein (CRP), procalcitonin (PCT), leukocytes, partial pressure of arterial oxygen (paO$_2$), and partial pressure of arterial carbon dioxide (paCO$_2$), thoracic imaging (chest radiography or computed tomography) and respiratory support (CPAP, non-invasive or invasive ventilation). All potentially collected parameters are described in Table 3. To check postoperative mental status, the Mini-Cog test will be used. This test consists of a 3-word recall task and the clock drawing test [21].

Additionally, after hospital discharge, relevant pulmonary imaging, diagnoses or complications are extracted from the discharge letter.

Postoperative recovery and patient satisfaction as a secondary outcome parameter are going to be evaluated with the Quality of Recovery-9 (QoR-9) questionnaire (Table 4) on day 1,3, and 7 [22]. The score ranges from 0 to 18, with a higher score indicating a better subjective recovery.

Other secondary outcome parameters, hospital length of stay and in-hospital mortality, will be determined by a chart review after hospital discharge.

## Data management

A FileMaker™ database is used to record postoperative outcomes. Database constraints avoid duplicate entries and values outside the valid range of clinical routine data. The ARISCAT score will be determined pre-operatively with the help of a paper-based checklist. A detailed description of the score is provided in "Background and rationale". The score has been validated in several European populations [15,23]. The primary endpoint, POPC, is defined according to the consensus definition of the StEP collaboration [1] and complemented by the definition of the EPCO task force [20]. The definition is described in detail in the chapter "Outcome". As mental status is part of the StEP criteria, a brief cognitive screening test (Mini-Cog test) will be performed regularly during the post-operative visits. The sensitivity of this test is 0.76–0.99, and the specificity is 0.83–0.93 [24]. As a secondary endpoint, the Quality-of-Recovery 9 questionnaire will be applied to assess subjective recovery after surgery. The test is highly sensitive ($0.92 \pm 0.01$) with a high negative predictive value ($0.93 \pm 0.01$) in a German study collective [22]. The tests and the questionnaire are detailed in the chapter "Outcome".

To promote participant retention, all outcome data will be assessed while the patient is still hospitalised. If a patient drops out, for example, because of withdrawal of consent or unexpected surgery rescheduling and therefore missed ultrasound, the study protocol will be stopped immediately, no further data will be collected, and already gathered data will be deleted. In case of patient discharge from the hospital before performing postoperative visits on days 3 and/or 7, the study protocol will continue, and the available data will be processed if the postoperative visit on day one is documented.

The data are collected and stored in different formats. The inclusion criteria as well as the ARISCAT score, are going to be collected paper-based in the pre-anaesthesia outpatient clinic. Lung ultrasound is performed with the SonoSite PX (Fujifilm SonoSite Inc., Bothell, Washington, USA). The imaging data is transferred to the hospital's internal PACS

**Table 2. Definition of postoperative pulmonary complications.**

| POPC | Definition/Diagnostic | Severity |
|------|----------------------|----------|
| Atelectasis | Detected on computed tomography OR chest radiography [1] | Mild: therapeutic supplemental oxygen <0.6 $FiO_2$<br>Moderate: therapeutic supplemental oxygen ≥0.6 $FiO_2$, requirement for high-flow nasal oxygen or both<br>Severe: unplanned non-invasive mechanical ventilation, CPAP or invasive mechanical ventilation requiring tracheal intubation<br>None: planned use of supplemental oxygen or mechanical respiratory support as part of routine care, but not in response to a complication or deteriorating physiology.<br>Therapies which are purely preventive or prophylactic<br>for example, high flow nasal oxygen or CPAP should be recorded as none |
| Pneumonia | US Centers for Disease Control Criteria: Two or more serial chest radiographs with at least one of the following (one radiograph is sufficient for patients with no underlying pulmonary or cardiac disease): (i) New or progressive and persistent infiltrates (ii) consolidation (iii) cavitation<br>AND at least one of the following: (a) fever (>38°C) with no other recognized cause (b) leukopenia (white cell count < $4x10^9$ litre-1) or leukocytosis (white cell count > $12x10^9$ litre-1) (c) for adults >70 years old, altered mental status with no other recognized cause<br>AND at least two of the following: (a) new onset of purulent sputum or change in character of sputum, or increased respiratory secretions, or increased suctioning requirements (b) new onset or worsening cough, or dyspnea, or tachypnoea (c) rales or bronchial breath sounds (d) worsening gas exchange (hypoxemia, increased oxygen requirement, increased ventilator demand) [1] | |
| Acute respiratory distress syndrome | Berlin consensus definition:<br>Timing: within 1 week of a known clinical insult or new or worsening respiratory symptoms<br>AND<br>Chest imaging: bilateral opacities not fully explained by effusions, lobar/lung collapse or nodules<br>AND<br>Origin of edema: respiratory failure not fully explained by cardiac failure or fluid overload (requires objective assessment, e.g., echocardiography, to exclude hydrostatic edema)<br>AND<br>Oxygenation:<br>mild $paO_2$:$FiO_2$ between 26.7 and 40.0 kPa (200–300 mmHg) with PEEP or CPAP ≥5 cmH2O<br>moderate $PaO_2$:$FiO_2$ between 13.3 and 26.6 kPa (100–200 mmHg) with PEEP ≥5 cmH2O<br>severe $PaO_2$:$FiO_2$ ≤ 13.3 kPa (100 mmHg) with PEEP ≥5 cmH2O.<br>OR<br>Mechanical ventilation<br>The need for tracheal re-intubation and mechanical ventilation after extubation, and within 30 days after surgery OR mechanical ventilation for more than 24h after surgery OR Non-invasive ventilation [1] | |
| Pulmonary aspiration | Clear clinical history (inhalation of regurgitated gastric contents) AND radiological evidence [1,14] | |
| Respiratory failure | Postoperative $PaO_2$ < 8 kPa (60 mmHg) on room air OR a $PaO_2$: $FiO_2$ < 40 kPa (300 mmHg)<br>OR arterial oxyhemoglobin saturation measured with pulse oximetry < 90% and requiring oxygen therapy [14] | n/a |
| Pulmonary embolism | Imaging finding (Computed tomography, ventilation/perfusion single photon emission computed tomography, echocardiography) | n/a |
| Pleural effusion | Chest radiograph with blunting of costophrenic angle, loss of sharp silhouette of the ipsilateral hemidiaphragm in upright position, displacement of adjacent anatomical structures, or (in supine position) hazy opacity in one hemithorax with preserved vascular shadow [14] | n/a |
| Pneumothorax | Air in the pleural space with no vascular bed surrounding the visceral pleura [14] | n/a |
| Bronchospasm | Newly detected expiratory wheezing treated with bronchodilators [14] | n/a |
| Cardiogenic pulmonary edema | Diffuse alveolar interstitial infiltrates with dyspnea and rales related to left ventricular failure, confirmed by echocardiography, pulmonary catheter or clinical improvement with specific treatment [15] | n/a |

Postoperative pulmonary complications are defined by the StEP collaboration [1] and complemented by EPCO criteria [20]. Severity is only defined for atelectasis, pneumonia, acute respiratory distress syndrome and aspiration. CPAP = continuous positive airway pressure, $FiO_2$ = fraction of inspired oxygen, $paO_2$ = partial pressure of arterial oxygen, POPC = Postoperative pulmonary complications.

**Table 3. Postoperative visit.**

| Category | Parameters |
|---|---|
| **Vital parameters** | |
| | Respiratory rate [1/min] |
| | Blood pressure (syst./diast.) [mmHg] |
| | Heart rate [1/min] |
| | Oxygen saturation [%] |
| | Temperature [°C] |
| | Oxygen supply [l/min] |
| **Laboratory** | |
| | Leukocyte count [giga/l] |
| | CRP [mg/l] |
| | PCT [μg/l] |
| **Pulmonal Symptoms** | |
| | Newly appeared coughing? (yes/no) |
| | Dyspnea? (yes/no) |
| | Newly appeared purulent sputum? (yes/no) |
| | Increased respiratory secretion? (yes/no) |
| | Increased suctioning requirements? (yes/no) |
| | Bronchial breath sounds (free text) |
| | Aspiration (yes/no) |
| **Blood gas analysis** | |
| | $paO_2$ [mmHg] |
| | $paCO_2$ [mmHg] |
| **Escalation of therapy** | |
| | Admission to PACU or ICU (yes/no) |
| | NIV (yes/no) |
| | CPAP (yes/no) |
| | Intubation (yes/no) |
| | $FiO_2$ [%] |
| | Antibiotics (free text) |
| | Dose of antibiotics (free text) |
| | Bronchodilating medication (free text) |
| | Dose of bronchodilating medication (free text) |
| **Thoracic imaging** | |
| | Chest radiography (free text) |
| | Computed tomography (free text) |
| **Mental status** | |
| | Overall score of the Mini-Cog test (3-word-recall-task and clock drawing test) |

Table 3 contains parameters that are collected on postoperative visits on day 1,3 and 7; CPAP = continuous positive airway pressure, CRP = C-reactive protein, diast. = diastolic, $FiO_2$ = fraction of inspired oxygen, ICU = intensive care unit, NIV = non-invasive ventilation, $paCO_2$ = partial pressure of arterial carbon dioxide, $paO_2$ = partial pressure of arterial oxygen, PACU = post-anaesthesia care unit, PCT = Procalcitonin, syst. = systolic.

and stored and processed in Digital Imaging and Communications in Medicine (DICOM) format within the Department of Radiology. Routinely collected perioperative data are stored in the hospital's internal patient data management system (PDMS) in an Oracle8i database. The data is accessible via Structured Query language (SQL) and can be exported in csv format. Postoperative visits are conducted in the form of structured questionnaires using tablets. Data collection and

**Table 4. Quality of recovery-9 questionnaire.**

| QoR-9-Questions | Score |
|---|---|
| Had a feeling of general well-being. | Not at all = 0 points<br>Some of the time = 1 point<br>Most of the time = 2 points |
| Had support from others (especially doctors and nurses). | |
| Been able to understand instructions and advise. Not being confused. | |
| Been able to look after personal toilet and hygiene unaided. | |
| Been able to pass urine and having no trouble with bowel function. | |
| Been able to breathe easily. | |
| Experiencing headache, backache or muscle pains | Not at all = 2 points<br>Some of the time = 1 point<br>Most of the time = 0 points |
| Experiencing nausea, dry-retching or vomiting | |
| Experiencing severe pain or constant moderate pain | |

The table was adapted from [22]; The article was published under the Creative Commons Attribution (CC BY 4.0) license (https://creativecommons.org/licenses/by/4.0/). QoR-9 = Quality of recovery-9.

processing are done with FileMaker Pro software (Claris, version 19.6.3.302). In the electronic data entry form for the postoperative visits, no user input is possible outside the valid value ranges for numerical data; dichotomous questions (yes/no) are documented with the help of radio buttons. The clock is drawn by the patient on the tablet and also stored in the database as a drawing. Together with the performance from the word-recall task, the score of the Mini-Cog test is calculated. Regarding the correct evaluation of the drawn clock, the score is cross-checked by a physician before it is finally transferred to the database. After patient discharge, discharge notes are searched for relevant diagnoses and complications. The total volume of data collected will be merged into a comprehensive FileMaker database. Data storage occurs pseudonymized and de-identified.

No laboratory evaluations or biological specimens outside the clinical routine will be obtained or stored as part of this study. For the detection of postoperative pulmonary complications, diagnostic tests and relevant results performed by the respective departments will be collected during a chart review and directly transferred to the pseudonymized data collection.

Any documents with identifiable information will be collected in paper-based form and stored in a locked cabinet at the study centre, where only authorised personnel will have access to them. This includes original informed consent, a checklist with eligibility criteria, the ARISCAT score, and the patient identification list. The documents will be kept there until completion of the study. The identification list will be stored separately, and only authorised study personnel will have access to it. After completion of the study, all paper records will be stored in a central archive for at least ten years according to the clinic's specifications and legal requirements. Information collected during the postoperative visits will be saved without any identifiable information on password-protected study tablets. After transferring the data to a hospital-internal computer, they are deleted from the tablets.

Pseudonymised data received from PDMS, chart review, and postoperative visits will be securely stored in the hospital's internal server infrastructure according to GDPR requirements. Imaging data is exclusively stored within the clinic's radiology information system.

## Statistical methods

The primary objective of the PEPPERMINT study is to develop and evaluate prediction models for postoperative pulmonary complications (POPC) using prospectively collected clinical and imaging data. Model development and validation will be conducted in accordance with the TRIPOD (Transparent Reporting of a Multivariable Prediction Model for Individual Prognosis or Diagnosis) guidelines. The statistical analysis plan is provided in S3 File.

## Overview of modelling approaches

Three predictive modelling strategies will be pursued. First, an imaging model will be created using deep learning techniques applied to lung ultrasound images. This model will be developed in Python using the PyTorch framework. Transfer learning will be employed, utilizing pretrained convolutional neural network (CNN) architectures such as ResNet or DenseNet. These models will be fine-tuned on the study-specific dataset. We will also assess the performance of medical foundation models and contrastive learning-based pretraining to optimize feature extraction from the ultrasound images.

Second, a clinical model will be developed using only structured patient data. This model will be trained using frameworks like H2O AutoML within R/RStudio as well as built-in R functions, which allows for the systematic evaluation and tuning of various machine learning algorithms, including gradient boosting machines, random forests, and neural networks.

Third, a combined model will integrate both clinical and imaging data. Various fusion strategies will be explored, including early and late fusion, to determine the most effective method for combining heterogeneous data types. The final architecture for this combined model will be selected based on performance observed in a hold-out test dataset.

## Model training and evaluation

The primary model development will be based on a dataset of 512 patients. Within this cohort, we will implement stratified k-fold cross-validation (typically 5- or 10-fold, depending on the final class distribution) where possible to assess model robustness and mitigate overfitting. Image-based deep learning models require extensive computational resources, therefore, cross-validation will be applied more selectively. We will therefore use a fixed training/validation/test split for imaging data, ensuring non-overlapping patient groups. Early model experiments will be conducted using internal validation on the training data with hold-out validation to optimize architecture and hyperparameters. Final model evaluation will be performed on an independent hold-out test set.

This hold-out test set will be compiled prospectively during the course of the study after model development, using newly enrolled patients not included in the training dataset. This temporal separation will ensure unbiased performance evaluation of the final model.

Model performance will be evaluated on the independent hold-out test set using several complementary performance metrics. These will include the area under the receiver operating characteristic curve (AUROC), the area under the precision-recall curve (AUPRC), overall accuracy, sensitivity, specificity, F1-score, as well as positive and negative predictive values. Calibration will be assessed using appropriate measures, such as the Brier score, Hosmer-Lemeshaw-Test and calibration curves. Clinical decision-making utility can additionally be evaluated using decision curve analysis (DCA).

Importantly, our models are designed to output individualized risk probabilities, not dichotomous predictions. As such, the selection of a cut-off point for potential clinical decision-making will be a post-modelling step, and risk thresholds for high-risk patient classification will not be predefined. Instead, various thresholding strategies will be explored post hoc, including Youden's index (for balanced sensitivity and specificity), cut-offs optimized for specific clinical priorities (e.g., maximizing sensitivity), and thresholds guided by DCA.

## Addressing overfitting and class imbalance

Overfitting will be addressed through the implementation of several regularization techniques commonly used in deep learning. These include dropout layers within the network architecture, L2 weight regularization to penalize large weights, and data augmentation techniques applied to ultrasound images (e.g., rotation, flipping, and zooming). Model complexity will also be restricted as needed.

To better estimate the prevalence and clinical relevance of POPC, we conducted a preliminary single-center observational study involving 259 patients undergoing surgery under general anesthesia. The cohort included 106 female patients (41%) and 62 current smokers (24%) with a median age of 66 years. Overall, 111 patients (43%) experienced at least one POPC, indicating that POPC is a relatively frequent complication in the targeted patient population [25].

Based on this preliminary observational data, only moderate class imbalance is to be expected. Nevertheless, additional strategies will be employed to ensure balanced learning. These include maintaining class ratios across cross-validation folds, monitoring class-wise performance metrics, and using class weighting techniques if imbalances are observed in specific subgroups.

## Missing values and data imputation

With regard to imaging data, only complete data sets will be accepted for analysis.

Missing values in the clinical dataset will initially be handled using the default preprocessing pipeline in the case that H2O AutoML is used. This pipeline applies median or mean imputation for numerical variables and mode imputation for categorical variables (https://docs.h2o.ai/h2o/latest-stable/h2o-docs/data-munging/imputing-data.html). In addition to this standard approach, we will evaluate alternative imputation strategies, particularly for approximately normally distributed numerical variables. These include, e.g., k-nearest neighbors (kNN) imputation, which leverages similarities across observations in the multivariate feature space and multiple imputation using chained equations (MICE), which incorporates multivariable relationships and reflects the uncertainty associated with imputed values.

Given the prospective nature of the study, we anticipate that most missing data will not be missing at random (NMAR), but rather occur systematically, for instance, due to early patient discharge or an inability to perform assessments as a result of clinical deterioration. In such cases, the absence of data may itself be informative with respect to the patient's risk for postoperative complications. Therefore, in addition to the imputation techniques mentioned above, we will explore modelling strategies that explicitly account for informative missingness. These include the incorporation of missingness indicators (binary flags denoting whether a variable was observed or missing) and sensitivity analyses to assess the robustness of model predictions under different assumptions about the missing data mechanism.

The final approach to handling missing data will be selected based on a combination of model performance, diagnostic checks, and clinical plausibility.

## Secondary models and endpoints

In addition to the prediction models, we aim to extract specific predictive markers which might be evaluated in an additional study. Once established and evaluated with additional studies, both the AI models and the marker might help to identify high-risk patients, allowing for the adaptation of treatment at an early point of care.

In addition to the primary binary classification model for the presence or absence of POPC, the study will explore the development of additional models that aim to distinguish between different types and severities of complications. These extended models will incorporate more granular outcome definitions, including the timing of complication onset and the presence of multiple complications in a single patient.

Moreover, secondary predictive models will be constructed to estimate outcomes such as quality of postoperative recovery, length of hospital stay, and in-hospital mortality. Important features will be identified based on variable importance metrics. Comparisons between patients with and without complications will be conducted using appropriate statistical tests, including t-tests or Mann-Whitney U tests for continuous variables, and chi-squared or Fisher's exact tests for categorical variables.

## External and subgroup validation

While model development and internal validation will be performed using data from a single clinical centre, further validation is planned at a second site of our hospital that predominantly treats gynaecological and ENT patients. This cohort will serve as a distinct population for assessing model generalisability across surgical disciplines.

Additionally, the model will be evaluated in a separate high-risk subgroup comprising patients undergoing urgent or emergency procedures, such as elderly individuals with hip fractures. These patients were intentionally excluded from the initial development cohort and will offer insight into the model's performance in more acute care settings.

## Interim analysis

An interim analysis will be conducted after the enrollment of approximately 50 patients (around 10% of the planned sample size). This analysis will assess the technical feasibility, data quality, and operational workflow. As this is a non-interventional study, early termination is not planned regardless of interim results.

## Oversight and monitoring

The Coordinating investigators are responsible for study design, funding, and creation of the study protocol. They take over the coordination and communication between the two involved departments, the Department of Anaesthesiology and Intensive Care Medicine and the Department of Radiology, and the persons involved in the study. Furthermore, they are part of the trial steering committee.

The trial steering committee is responsible for adhering to the study protocol, conducting the planned patient enrollment, and compiling the patient identification list. They will monitor the study's progress and, if necessary, agree on changes to the procedure and study protocol. The committee meets once every 2 months in an in-person or online meeting.

The Lead investigator is responsible for eligibility, consent, and enrollment of patients as well as imaging and data collection, and therefore supports the practising physicians on a day-to-day basis.

The present study is not a study according to the German Medical Product Act (AMG) or the Medical Devices Regulation (MDR). The study is monocentric, and no intervention will take place. No risks for the patients are to be expected. For these reasons, no external data monitoring committee will be set up.

However, the coordinating investigators will be responsible for the creation of the database, supporting data entry, data verification, and quality management. Data monitoring and outcome reports will take place every 8 weeks.

## Adverse events and harms

Patients will undergo routine perioperative care as per standard during this trial; responsible for patient care will be the attending physicians and departments. Additionally, patients will receive (1) a lung ultrasound, considered non-invasive and without side effects, and (2) a clinical examination and interview at up to three time points postoperatively. All other procedures are part of general anaesthesia or usual perioperative management and are completed even without study participation. Therefore, we do not expect any complications or harm from trial participation.

Nevertheless, patients can report adverse events or other unintended effects of the trial to the study hotline or email address. The trial steering committee will process the reports.

Due to the relatively small size of the data set, the prediction model will be developed using the whole dataset. Cross-validation will be used to evaluate the performance of the model. Furthermore, validation is planned on a temporally independent data set obtained during the period of model training. Therefore, the study will be classified as Type 1b according to the TRIPOD Statement [12].

Error analysis will be carried out with the help of a confusion matrix after threshold determination. The cases of incorrect predictions will be analyzed in more detail, in particular, to determine whether certain characteristics correlate with the errors. External validation will take place in future studies.

In the current study, we are not focusing on the identification of possible confounders for model performance. This will be done in a follow-up study, investigating the impact of possible confounders such as different raters, imaging devices, bad image quality, resolutions, etc., in a possibly multi-centric study.

## Discussion

A major target of clinical research in the perioperative field is to reduce the occurrence of postoperative complications. In times of skills and resource shortage, personalized medicine is getting more important, which includes the application of

required treatment but avoids overtreatment. Machine learning algorithms might improve risk prediction as a prerequisite for personalised medicine. POPC represent a large proportion of the overall postoperative complications and occur about twice as frequently as cardiac complications. POPC are not only common, but they are also responsible for increased morbidity and mortality. Furthermore, they contribute to increased hospital length of stay and a higher frequency of hospital readmissions. Therefore, they occupy more healthcare resources and cause higher healthcare costs [26–29].

Despite these facts, there are only a few scores for evaluating pulmonary risk, which have not yet become standards in clinical routine, even though pulmonary complications could be controlled and avoided by specific, however, personnel-intensive measures.

Lung ultrasound is a non-invasive, bedside diagnostic screening measure that has recently become increasingly popular, not at least due to the COVID-19 pandemic. Standardised protocols and guidelines mean lung ultrasound is becoming increasingly important in clinical medicine [30]. Increasingly, machine-learning models based on ultrasound examinations are being developed that deliver high diagnostic accuracy [31] and already exceed the currently best evaluated conventional score, the ARISCAT Score [32].

The PEPPERMINT study aims to develop a tailored machine-learning model to reliably predict the risk for POPC, based on lung ultrasound imaging performed in the recovery room and perioperatively assessed clinical data. We hypothesize that the accuracy of the prediction model outperforms common scoring systems. Early identification of patients at risk helps to target scarce resources and apply adequate therapy in the sense of personalised medicine.

## Limitations

We wanted to set a specific time frame for the post-operative visits. Therefore, in-person visits occur exclusively during the first 7 postoperative days. After that timepoint, the survey of findings is limited to a chart review. However, since the majority of POPC occurs within the first week [26], this pragmatic approach is justified.

Secondly, risk identification by the model will take place only after the surgery. Therefore, preoperative assessment and optimization are not the subject of our study. However, if one considers the criteria that are relevant in preoperative risk assessment scores [5], for example, age, respiratory infection, or expected surgery duration and incision, most of the criteria are related to the underlying disease or planned surgical procedure, and are therefore not amenable to preoperative modifications. Consequently, in this study we would like to develop a tool that reliably predicts complications in the early postoperative phase in order to be able to provide the patient with adequate postoperative treatment and monitoring.

## Strengths

The PEPPERMINT study will be the first study to combine ultrasound imaging data with clinical data in an artificial intelligence prediction model. We, therefore, hope to achieve a highly accurate risk prediction that can be applied in clinical practice. Besides POPC, we also investigate secondary endpoints that are of interest to the healthcare system, like the length of in-hospital stay and endpoints that are relevant to the subjective feelings of patients, like the quality of recovery.

In perspective, the suitability of the algorithm will be tested in a clinical intervention study. Therefore, a higher number of patients will be screened with the created model. High-risk patients receive a multimodal training and therapy program postoperatively to reduce the rate of POPC. This includes, among other things, non-invasive ventilation in the recovery room, physiotherapy, respiratory training, a nutrition plan to prevent malnutrition, fluid balancing to prevent overhydration, and special oral hygiene. All included patients will be again visited on the ward on days 1,3, and 7 and examined for signs of pulmonary complications. The aim is a reduction of pulmonary complications with measurable clinical benefit. Clinically measurable success parameters are a shorter hospital stay, a lower rate of unplanned intensive care admissions, and a higher quality of life.

Precise risk assessment using a machine-learning algorithm combined with targeted preventive and therapeutic measures for identified high-risk patients, therefore, has great potential to improve patient outcomes and could also help to reduce health care costs.

## Dissemination plans

Trial results will be communicated via publication in international, peer-reviewed journals and at international congresses in the fields of anaesthesia and radiology. Positive as well as negative results will be published.

## Protocol amendments

All important protocol modifications will be communicated to the necessary parties through the trial steering committee via direct contact or online meeting. Necessary changes in trial registries and ethics committee will be carried out as soon as possible.

## Trial sponsor

Prof. Dr. Bettina Jungwirth
University Hospital Ulm , Albert-Einstein-Allee 23 , 89081 Ulm
mail: ains@uniklinik-ulm.de
Prof. Dr. Meinrad Beer
University Hospital Ulm, Albert-Einstein-Allee 23 , 89081 Ulm
mail: sekretariat.radiologie1@uniklinik-ulm.de
This is an investigator-initiated trial. The funding source had no role in the design of this study and will not have any role during its execution, analyses, interpretation of the data, or decision to submit results.

## Trial status

- Protocol version: 2
- Issue Date: 21/07/2025
- Protocol Amendment Number: 0
- First day of recruitment: 25/04/2023
- Included patients: 150
- Approximate date of completed recruitment: 12/2025

## Supporting information

**S1 File. Checklist PEPPERMINT study.**
(PDF)

**S2 File. SPIRIT-AI checklist.** Recommended items to address in a protocol and related documents for clinical trials evaluating AI interventions.
(PDF)

**S3 File. Statistical analysis plan.**
(PDF)

## Author contributions

**Conceptualization:** Britta Trautwein, Meinrad Beer, Manfred Blobner, Bettina Jungwirth, Simone Maria Kagerbauer, Michael Götz.

**Data curation:** Simone Maria Kagerbauer, Michael Götz.

**Funding acquisition:** Simone Maria Kagerbauer, Michael Götz.

**Methodology:** Simone Maria Kagerbauer, Michael Götz.

**Supervision:** Manfred Blobner, Bettina Jungwirth.

**Visualization:** Britta Trautwein.

**Writing – original draft:** Britta Trautwein.

**Writing – review & editing:** Meinrad Beer, Manfred Blobner, Bettina Jungwirth, Simone Maria Kagerbauer, Michael Götz.

## Acknowledgments

We acknowledge F. Scheffenbichler, B. Ulm and A. Podtschaske for their support and advice in the implementation of the study. We acknowledge K. Lukas-Jazwinski, S. Hoheisen, F. Branz, G. Frömmichen, P. Leibinger and P. S. Sam for data acquisition and H. Hillenhagen and T. Bader for the preliminary work on the machine learning model. We also want to thank the patients for their willingness to participate in this study.

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
