## [Decision Letter · Decision Letter 0]

PONE-D-24-22158Preventing postoperative pulmonary complications by establishing a machine-learning assisted approach (PEPPERMINT): study protocol for the creation of a risk prediction modelPLOS ONE

Dear Dr. Trautwein,

Thank you for submitting your manuscript to PLOS ONE. After careful consideration, we feel that it has merit but does not fully meet PLOS ONE’s publication criteria as it currently stands. Therefore, we invite you to submit a revised version of the manuscript that addresses the points raised during the review process.

**ACADEMIC EDITOR: **

please carefully  assess all reviewers comments

We look forward to receiving your revised manuscript.

Kind regards,

Silvia Fiorelli

Academic Editor

PLOS ONE

Journal Requirements:

[I have read the journal's policy and the authors of this manuscript have the following competing interests: B. Jungwirth and S. Kagerbauer received grants from Löwenstein Medical Innovation (Berlin, Germany). M. Blobner received research support from MSD (Haar, Germany), fees for consultancy or lectures from GE Healthcare (Helsinki, Finland), Grünenthal (Aachen, Germany), and Senzime (Landshut, Germany).].

Please confirm that this does not alter your adherence to all PLOS ONE policies on sharing data and materials, by including the following statement: ""This does not alter our adherence to PLOS ONE policies on sharing data and materials.” (as detailed online in our guide for authors

http://journals.plos.org/plosone/s/competing-interests ). If there are restrictions on sharing of data and/or materials, please state these. Please note that we cannot proceed with consideration of your article until this information has been declared.

3. In the online submission form, you indicated that [Due to German data protection regulations, we are not allowed to publish individual patient data. Access to the full protocol and data can be shared upon reasonable request from the corresponding author after appraisal by the data protection officer. The statistical code and machine learning code will be shared via a GitHub repository.]

Reviewers' comments:

Reviewer's Responses to Questions

**Comments to the Author**

1. Does the manuscript provide a valid rationale for the proposed study, with clearly identified and justified research questions?

Reviewer #1: Partly

Reviewer #2: Yes

2. Is the protocol technically sound and planned in a manner that will lead to a meaningful outcome and allow testing the stated hypotheses?

Reviewer #1: Yes

Reviewer #2: Yes

3. Is the methodology feasible and described in sufficient detail to allow the work to be replicable?

Reviewer #1: Yes

Reviewer #2: Yes

4. Have the authors described where all data underlying the findings will be made available when the study is complete?

Reviewer #1: No

Reviewer #2: Yes

5. Is the manuscript presented in an intelligible fashion and written in standard English?

Reviewer #1: Yes

Reviewer #2: Yes

6. Review Comments to the Author

You may also provide optional suggestions and comments to authors that they might find helpful in planning their study.

Reviewer #1: The process of physicians deciding whether images meet criteria and potentially excluding patients introduces selection bias.

Using mean or modal imputation for missing data is overly simplistic and can introduce bias. More sophisticated methods like multiple imputation may be considered.

What test was used for the sample size calculation?

While AUROC is mentioned, in imbalanced datasets (which are common in medical studies), AUPRC may be more informative.

The use of complex deep learning models with relatively limited data raises concerns about overfitting, especially with no clear regularization strategy mentioned.

There are no plans for cross-validation or external validation.

Given that POPC is likely a relatively rare event, there's no clear strategy for dealing with potential class imbalance in the dataset.

Using Youden's index for cut-off determination may not be optimal, especially if the costs of false positives and false negatives are not equal in this clinical context. How will the risk score be calculated?

Reviewer #2: This experimental protocol proposes developing a predictive model for post-surgical pulmonary complications by integrating standardized lung ultrasound examinations with clinical data and employing machine learning techniques based on neural networks and integrated methods. The approach is highly innovative, particularly for its potential application in the perioperative period—a complex clinical environment. It is convenient and cost-effective, with the benefit of serving as an early warning system in clinical practice. Below are some of my comments:

1. The core element in developing a disease risk prediction model using AI and machine learning is the algorithm itself. Developing such a model requires expertise in machine learning. Please provide more details about the professionals involved in your team, the design of the model, and the software tools utilized.

2. Has the model been externally validated? How do you plan to ensure its broad applicability across diverse clinical settings?

3. Lung ultrasound has limited use in clinical practice due to the interference of air in the lungs. Additionally, the complexity of the ultrasound images is influenced by factors such as probe position and angle. How to ensure the accuracy and reliability of the model considering these challenges?

4. Pulmonary complications can be presented in a variety of ways, with differing severity. Consequently, there will likely be numerous outcome events (endpoints) in this study. Some mild postoperative pulmonary complications may not require intervention, and predicting such cases may not provide substantial clinical value. Does this study aim to focus specifically on identifying the more severe pulmonary complications that necessitate medical intervention?

7. PLOS authors have the option to publish the peer review history of their article (what does this mean? ). If published, this will include your full peer review and any attached files.

**Do you want your identity to be public for this peer review?** For information about this choice, including consent withdrawal, please see our Privacy Policy .

Reviewer #1: No

Reviewer #2: No

---

## [Author Response · Author response to Decision Letter 1]

11 Jun 2025

We would like to thank the reviewers and editors for their thoughtful and constructive comments, which have helped us to substantially improve our manuscript.

All reviewer comments were addressed in the “Response to reviewer” document and changed accordingly in the manuscript.

In response to the reviewers’ feedback, we have made several important revisions. Most notably, we conducted an additional observational study to determine the exact prevalence of postoperative pulmonary complications in our patient population. Furthermore, we have considerably expanded the section “Statistical analyses” in the manuscript to provide greater transparency and methodological clarity. As this section now covers the content of the original statistical analysis plan in sufficient detail, we have decided to omit the separate supplement containing the statistical analysis plan.

---

## [Decision Letter · Decision Letter 1]

Preventing postoperative pulmonary complications by establishing a machine-learning assisted approach (PEPPERMINT): study protocol for the creation of a risk prediction model

PONE-D-24-22158R1

Dear Dr. Britta Trautwein,

We’re pleased to inform you that your manuscript has been judged scientifically suitable for publication and will be formally accepted for publication once it meets all outstanding technical requirements.

Kind regards,

Silvia Fiorelli

Academic Editor

PLOS ONE

Additional Editor Comments (optional):

Congratulations to the authors and thanks to the reviewers for the provided suggestions which really helped improve the quality of the manuscript

Reviewers' comments:

Reviewer's Responses to Questions

**Comments to the Author**

1. Does the manuscript provide a valid rationale for the proposed study, with clearly identified and justified research questions?

Reviewer #1: Yes

2. Is the protocol technically sound and planned in a manner that will lead to a meaningful outcome and allow testing the stated hypotheses?

Reviewer #1: Yes

3. Is the methodology feasible and described in sufficient detail to allow the work to be replicable?

Reviewer #1: Yes

4. Have the authors described where all data underlying the findings will be made available when the study is complete?

Reviewer #1: Yes

5. Is the manuscript presented in an intelligible fashion and written in standard English?

Reviewer #1: Yes

6. Review Comments to the Author

You may also provide optional suggestions and comments to authors that they might find helpful in planning their study.

Reviewer #1: All my concerns are addressed.

7. PLOS authors have the option to publish the peer review history of their article (what does this mean? ). If published, this will include your full peer review and any attached files.

**Do you want your identity to be public for this peer review?** For information about this choice, including consent withdrawal, please see our Privacy Policy .

Reviewer #1: No

---

## [Editor Report · Acceptance letter]

PONE-D-24-22158R1

PLOS ONE

Dear Dr. Trautwein,

I'm pleased to inform you that your manuscript has been deemed suitable for publication in PLOS ONE. Congratulations! Your manuscript is now being handed over to our production team.

Kind regards,

on behalf of

Dr. Silvia Fiorelli

Academic Editor

PLOS ONE